# Impact of Maternal *Moringa oleifera* Leaf Supplementation on Milk and Serum Vitamin A and Carotenoid Concentrations in a Cohort of Breastfeeding Kenyan Women and Their Infants

**DOI:** 10.3390/nu16193425

**Published:** 2024-10-09

**Authors:** Suzanna Labib Attia, Silvia A. Odhiambo, Jerusha N. Mogaka, Raphael Ondondo, Aric Schadler, Kristen McQuerry, George J. Fuchs, Janet E. Williams, Michelle K. McGuire, Carrie Waterman, Kerry Schulze, Patrick M. Owuor

**Affiliations:** 1Department of Pediatrics, University of Kentucky, Lexington, KY 40506, USA; schadler@uky.edu (A.S.); george.fuchs@uky.edu (G.J.F.III); 2Pamoja Community Based Organization, Kisumu 2311-40100, Kenya; silvia@pamojapamoja.org (S.A.O.); owuor@wayne.edu (P.M.O.); 3School of Public Health, University of Alberta, Edmonton, AB T6G 2E2, Canada; jerusha7@uw.edu; 4School of Nursing, University of Washington, Seattle, WA 98195, USA; 5Masinde Muliro University of Science and Technology, Kakamega P.O. Box 190-50100, Kenya; raphondondo@gmail.com; 6Department of Biostatistics, College of Public Health, University of Kentucky, Lexington, KY 40506, USA; kristen.mcquerry@uky.edu; 7Department of Animal, Veterinary and Food Sciences, University of Idaho, Moscow, ID 83844, USA; janetw@uidaho.edu; 8Margaret Ritchie School of Family and Consumer Sciences, University of Idaho, Moscow, ID 83844, USA; smcguire@uidaho.edu; 9Institute for Global Nutrition, University of California Davis, Davis, CA 95616, USA; carriewaterman22@gmail.com; 10Department of International Health, Johns Hopkins Bloomberg School of Public Health, Baltimore, MD 21205, USA; kschulz1@jhu.edu; 11Department of Anthropology, Wayne State University, Detroit, MI 48202, USA

**Keywords:** human milk, vitamin A, maternal and child health, carotenoids

## Abstract

**Background:** Childhood vitamin A deficiency leads to increased morbidity and mortality. Human milk is the only source of vitamin A for exclusively breastfed infants. Dried *Moringa oleifera* leaf powder (moringa) is a good food source of provitamin A and other carotenoids. Its effect during lactation on human milk vitamin A and carotenoid content is unclear. **Objectives:** Our objective was to investigate the effect of maternal moringa consumption on human milk retinol and carotenoid concentrations and maternal and infant vitamin A status. Methods: We conducted a 3-month pilot single-blinded cluster-randomized controlled trial in breastfeeding mother–infant pairs (n = 50) in Kenya. Mothers received corn porridge with (20 g/d) or without moringa with complete breast expressions and maternal and infant serum collected at enrollment (infant <30 days old) and 3 months. Milk was analyzed for retinol and selected carotenoids; maternal/infant serum was analyzed for retinol binding protein (RBP). **Results:** 88% (n = 44) pairs completed milk and serum samples. Four mothers (9%) had vitamin A deficiency (RBP <0.07 µmol/L); 11 (25%) were vitamin A insufficient (VAI; RBP <1.05 µmol/L). Alpha-carotene concentration in milk was higher in the moringa than the control group at baseline (*p* = 0.024) and at exit (least squares means, LSM, 95%CI µg/mL 0.003, 0.003–0.004 moringa vs. 0.002, 0.001–0.003 control, n = 22/cluster; *p* = 0.014). In mothers with VAI, alpha-carotene was higher in the moringa group than controls at exit (LSM, 95%CI µg/mL 0.005, 0.003–0.009 moringa, n = 3, vs. 0.002, 0.000–0.004 control, n = 8, *p* = 0.027) with no difference at baseline. Milk carotenoids did not correlate with vitamin A status (serum RBP) in infants or mothers. **Conclusions:** Maternal moringa consumption did not impact concentration of milk vitamin A and resulted in limited increase in milk carotenoids in this cohort.

## 1. Introduction

Vitamin A deficiency (VAD) affects up to 29% of children under five years in low- and middle-income countries and is one of the most common nutritional disorders globally [1]. VAD leads to significant morbidity and mortality, including an increased rate of infection and childhood death from infection, and is the leading cause of preventable childhood blindness [2,3,4,5]. Human milk is the only source of vitamin A for exclusively breastfed infants, and, thus, maternal vitamin A deficiency can also lead to vitamin A deficiency in breastfed infants [3,6,7,8,9].

Vitamin A through milk is provided to the infant in the form of provitamin A carotenoids and preformed vitamin A is provided in the form of retinol; these are influenced by the maternal diet [10]. Carotenoids are a group of dietary compounds that cannot be synthesized de novo by the human body and perform key functions in the growing infant. Beta-carotene is a provitamin A carotenoid that is cleaved to retinol in the intestine when vitamin A status is poor, but it is also absorbed in relation to intake of beta-carotene-rich food sources and functions as an antioxidant. Though beta-carotene is the most efficiently utilized provitamin A carotenoid, alpha-carotene and beta-cryptoxanthin are also potential sources of vitamin A. Lycopene is a non-provitamin A carotenoid with strong antioxidant activities [11]. Lutein accumulates in the eye and brain in the growing infant and is key to adequate vision and neurocognitive development along with its isomer zeaxanthin; it additionally functions as an antioxidant [12]. Recommendations for post-partum supplementation of vitamin A in lactating mothers and are no longer routinely recommended by the World Health Organization. Some countries, however, still provide varying amounts up to 400,000 IU total in the postpartum period (120,000 µg RAE) [13,14].

*Moringa oleifera* is a rapidly growing and nutrient-dense tree which is rich in carotenoids [15]. Varieties of the moringa family (Moringaceae) are widespread globally and have been used as food and traditional medicine for centuries [16]. The most researched, *Moringa oleifera,* is native to the Himalayas and cultivatable in a variety of climates, including arid, drought-prone regions [17]. The leaves are increasingly the focus of research due to their high nutrient content and abundant anti-inflammatory compounds, most notably isothiocyanates. The leaves are ready for harvest throughout the year [18] and are easily dried and powdered for shelf-stable use with little economic or technological investment [17]. Thus, moringa is a portable, highly nutritious food source suitable for use in limited-resource populations, especially in low and middle-income countries where food insecurity and undernutrition are exacerbated by climate change, displacement, or conflict [17].

Moringa leaves have high concentrations of protein, iron, B vitamins, calcium, and other essential nutrients as well as fiber [15,19]. Moringa leaves contain high amounts of provitamin A carotenoids, especially beta-carotene and beta-cryptoxanthin, which some studies estimate as up to 36.4 µg of retinol activity equivalents (RAE) per gram of dried moringa leaf powder [20,21]. Moringa is an excellent source of other carotenoids as well, such as lutein, with documented concentrations of up to 204 µg/g in fresh moringa leaves [19,22]. Lycopene content has been estimated at nearly 10 mg/g in moringa leaf powder [23].

Moringa leaf supplementation at 14 g/day has been shown to improve vitamin A status in adult women with vitamin A deficiency [24]. The effect of moringa supplementation on human milk carotenoids remains unknown. The primary aim of this study was to document the role of maternal moringa supplementation on concentrations of selected carotenoids in human milk in a limited-resource setting among mothers of full-term breastfeeding infants. We hypothesized that moringa supplementation would lead to improvements in maternal milk retinol, provitamin A carotenoids, lutein, and lycopene and may be associated with improved maternal and infant vitamin A status (serum retinol binding protein, RBP).

## 2. Methods

This study was part of a previously described parent pilot study designed to investigate the impact of maternal moringa supplementation on infant growth, micronutrient deficiency, and milk output and composition [25].

### 2.1. Experimental Design

We performed a single-blinded cluster-randomized controlled trial of 20 g/d of moringa leaf powder in corn porridge vs. corn porridge alone taken by lactating mothers of term infants as 10 g leaf powder in 1 cup of corn porridge twice daily or 1 cup of corn porridge without moringa twice daily for three months.

### 2.2. Inclusion/Exclusion Criteria

Exclusively breastfeeding women (≥18 years) and their <30-day old infants born singleton and full-term (≥36 weeks gestation) were eligible to participate. Mothers were required to be able to eat by mouth, demonstrate fluency in Luo, Swahili, or English, and were exclusively breastfeeding and planned to continue for the duration of the study. In addition, women with contraindications for lactation, consuming moringa regularly (>2 times/week in the prior month), receiving regular fortified food supplementation, refusing to consume corn porridge with or without moringa at least 3 times within the first study month, or who chose to withdraw or were lost to follow-up within the first 2 study weeks were excluded. To be enrolled in the study, infants were required to be without significant congenital disease affecting growth or ability to orally feed (ex. trisomy 13 or significant heart disease) and able to breastfeed. HIV status was not a criterion for exclusion in mothers or infants.

### 2.3. Location

Study activities were conducted in western Kenya at the Chulaimbo Sub-County Hospital and Kombewa Sub-County Hospital in Kisumu County. These two hospitals were chosen because they offer maternal and child health services, have laboratory services, and serve a diverse population. Kisumu County is the catchment region of study partner Pamoja Community Based Organization (CBO), a grassroots non-profit with a mission to identify and address the most salient needs affecting the wellbeing of families, especially women and children, in Kisumu County. Over 70% of the population of Kisumu County live below the poverty level and childhood malnutrition is common (stunting 9.1%, wasting 3%, and underweight 3.5% in children <5 years old as of 2022) [26].

### 2.4. Ethical Review

This study is registered at ClinicalTrials.gov (Clinical trials registration: https://classic.clinicaltrials.gov/ct2/show/NCT05333939). Ethical approval was obtained from the University of Kentucky Medical Institutional Review Board (#58219, 23 September 2020) and Amref Health Africa Ethics and Scientific Review Committee in Kenya (#P923-2020, 21 April 2021). Written informed consent was obtained from each participating mother in her preferred language (Luo, Swahili, or English).

### 2.5. Dietary Intervention

The intervention was tested first through an acceptability study. Participants (mothers) received 20 g of moringa leaf powder divided as 10 g of moringa in 1 cup of corn porridge consumed twice daily if in the intervention group. The corn porridge was made of corn flour boiled with water. Participants in the control group consumed 1 cup of corn porridge twice daily. All participants were allowed to add salt or sugar to taste; these were not provided. This was provided on top of mothers’ ad lib regular diet. Corn flour and moringa leaf powder (intervention group only) were delivered monthly to mothers. At each month follow-up, remaining corn and moringa flour were weighed. Mothers received the same measuring spoon and cup for measurement of corn and/or moringa flour. Mothers were instructed to record daily consumption of corn porridge with or without moringa in a diary using pictorial (if numerically illiterate) or numerical representation of volume. The diaries were reviewed at each monthly follow-up. Mothers were also asked at each visit if anyone else consumed the corn and/or moringa flour. We were unable to collect additional dietary data.

### 2.6. Milk and Serum Collection

Trained study staff collected milk, and phlebotomists collected serum samples at enrollment and three months. Milk samples were collected by hand expression by the mother assisted by the study staff and taken from whole breast samples at least 2 h after feeding. Milk samples were light protected, stored on dry ice, then at −20 °C, then at −80 °C until analysis. Serum samples were taken by obtaining capillary blood (finger for mother and heel for infant with up to 2 mL collected/sample), centrifugation of the whole blood, and then freezing of serum until analysis for RBP by VitMin Lab [27].

### 2.7. Sample Analysis

Milk retinol and carotenoid analysis was performed by high-performance liquid chromatography by a commercial laboratory, Eurofins, for retinol, lutein, zeaxanthin, alpha-cryptoxanthin, beta-cryptoxanthin, lycopene, cis-lycopene, alpha-carotene, provitamin A as beta-carotene, and cis-beta-carotene. Retinol activity equivalents (RAE) were calculated using the formula:

retinol + (β-carotene/12) + (α-carotene/24) + (β-cryptoxanthin/24), using least squares means levels (LSM) (µg/mL) [28].

### 2.8. Statistical Analysis

Data were analyzed using SAS (version 9.4, SAS Institute, Cary, NC, USA). The significance level was set at α = 0.05. Analysis was performed to account for the cluster-randomized design. Treatment was completely confounded with cluster. In order to address differences within clusters, i.e., accounting for the inter-cluster correlation, the mother–infant pair within treatment was used as the error term for all treatment hypothesis testing. Natural log transformations were performed for all retinol and carotenoid values and back-transformed estimates are presented. Associations with treatment and baseline characteristics were analyzed using generalized linear mixed models (GLMMs) with mother–infant pair within location considered a random effect. For binary responses, the logit link function was used with the mother–infant pair within location considered a random effect. For each carotenoid and timepoint (baseline/3 months), location differences were analyzed using GLMMs similar to the analysis used for baseline characteristics analysis. In order to assess the relationship of milk retinol and carotenoids to vitamin A status, subgroup carotenoid analyses were performed for low RBP at baseline and 3 months, and insufficient or sufficient RBP at baseline and 3 months. We were unable to analyze the relationship among treatment and time due to insufficient degrees of freedom (n = 2 clusters). For each carotenoid or retinol, RBP, location, and timepoint combination (baseline/3 months), Pearson correlation coefficients were computed. For each carotenoid or retinol and location, clinical outcome differences were analyzed using one-way ANOVA. It is important to note that for both the Pearson correlation coefficient and ANOVA analyses, independence was assumed within a cluster.

## 3. Results

### 3.1. General

A total of 88% (44 of 50) mother–infant pairs completed milk and serum sample collection and 3-month follow up. There were no differences in baseline demographics between groups of women completing the study (Table 1). HIV status was evenly distributed between groups. Maternal malnutrition, defined as mid-upper-arm circumference <23 cm, trended higher in the treatment group (18% vs. 5%, *p* = 0.199); however, birthweight and infant anthropometrics were normal overall (Table 1). Within the cohort as a whole, moringa leaf powder use was 11.8 ± 3.8 g/day between months 1 and 2 and 14.9 ± 3.4 g/day between months 2 and 3 of the 3-month study. Most infants (100% of the treatment group and 82% of the control group) were delivered by spontaneous vaginal delivery. Information on attendance at delivery by a birth professional was not collected. No mothers received prenatal vitamin A supplementation, nor did any mothers or infants receive postnatal vitamin A supplementation.

RBP as a proxy of vitamin A status and prevalence of vitamin A insufficiency and deficiency was similar between groups at baseline and exit for mothers and infants (Table 2).

### 3.2. Impact of Moringa on Milk Composition at Baseline and 3 Months by Treatment Group

Alpha-carotene concentration in milk was slightly higher in the moringa over the control group at 3 months (LSM, 95% CI µg/mL 0.003, 0.003–0.004 moringa vs. 0.002, 0.001–0.003 control, n = 22/cluster; *p* = 0.014; (Table 3)). Alpha-carotene was also higher at baseline in the moringa group (Table 3), and this difference remained statistically significant by 3 months (*p* = 0.024 at baseline, *p* = 0.014 at 3 months). Milk carotenoids did not correlate with serum RBP in infants or mothers. RAE showed similar values to LSM retinol and no differences between groups, as the contributions of alpha- and beta-carotene and beta-cryptoxanthin were quite small (Table 3).

### 3.3. Relationship of Milk Carotenoids and Retinol to Vitamin A Status

Vitamin A deficiency (<0.07 µmol/L RBP) was present in only four mothers (9.1%) at baseline and distributed evenly between groups (n = 2/cluster, Table 2). Vitamin A insufficiency (<1.05 µmol/L RBP, VAI) was present in 11 mothers at baseline (8 in the control group and 3 in the moringa group, Table 2). In those with VAI at baseline, beta-cryptoxanthin, alpha-cryptoxanthin, lutein, zeaxanthin, and total reported carotenoids were higher at baseline in the moringa group. By study end, these differences persisted for beta-cryptoxanthin; alpha-carotene by study end was higher in the moringa group than in controls (Table 4). VAI persisted in only 3 mothers by study end (2 in control and 1 in the moringa group). These numbers are considered too small to allow for analysis. Vitamin A sufficiency (≥1.05 µmol/L RBP, VAS) trended higher in the moringa group with VAS in 64% of mothers in the control group and 83% of mothers in the moringa group; this nonsignificant difference lessened by study end (Table 2). For mothers with VAS at baseline, there were no differences in retinol or carotenoids at baseline. At study end, beta-cryptoxanthin was higher in the control group than in the moringa group (LSM, 95 CI µg/mL 0.024, 0.014–0.034 control, n = 14 vs. 0.008, −0.000–0.017 moringa, n = 18, *p* = 0.018, Table 5). Changes in milk retinol and carotenoids over time with and without moringa supplementation according to vitamin A sufficiency or insufficiency status are presented in Figure 1.

Trends of milk carotenoid and retinol shifts related to infants with inadequate vitamin A levels reflected differences only in alpha-carotene. VAD was present in 27 infants at baseline (16 in control and 11 in moringa group) and VAI was present in 38 infants (21 in control and 19 in moringa group). The only notable difference was the persistence of higher milk alpha-carotene at study end in mothers of VAD or VAI infants.

## 4. Discussion

Until this study, the effect of moringa on human milk retinol and carotenoids was unknown. We observed that moringa supplementation in our population of Western Kenyan mothers and their term infants was associated with higher milk alpha-carotene levels over the control group regardless of vitamin A status. The difference in alpha-carotene may be to some extent driven by the higher baseline level in mothers receiving moringa supplementation. We were unable to correct for this due to the limited degrees of freedom with only n = 2 clusters in this pilot study. The number of mothers with VAD was too small to draw conclusions about the role of moringa supplementation in maternal VAD. Of note, 20 g of moringa leaf powder daily for 90 days would provide approximately 65,000 µg RAE total. This is lower than the maximum dose of 400,000 IU/120,000 µg RAE previously recommended by the WHO and still in use by some countries [13,14].

### 4.1. Comparison

Milk nutrients including carotenoids and retinol have been shown to decline over time in the first months post-partum [29]. This was true for our population as well. Although moringa leaves are a rich source of beta-cryptoxanthin, other nutrients which were expected to improve (lycopene, beta-carotene) did not appear to improve with moringa. Supplementation with select carotenoids, especially beta-carotene, has shown mixed results in the ability to change maternal milk carotenoids [27,28]. Canfield et al. showed a measurable increase in milk beta-carotene with a one-time 60 and 210 mg oral dose of beta-carotene [30]. As a single dose of dried leaf powder daily, we estimate that we were providing a much smaller amount of beta-carotene; however, this was provided longitudinally over three months [31].

Other studies using moringa leaf supplementation have shown improvements in vitamin A status in humans and improvements in milk and serum carotenoids in animals [23,32,33]. Animal studies of maternal moringa supplementation during lactation more specifically demonstrate increased carotenoid milk content with moringa supplementation vs. controls. Afzal et al. reported that goats supplemented with 3.5% of diet as moringa leaf powder (they note 13.87 ± 0.33 mg/g carotenoid content) for 28 days had total milk carotenoids of 3.04 ± 0.08 µg/mL vs. 1.38 ± 0.38 in controls at 28 days, *p* = 0.025, n = 10/group); they also reported improvements in total serum carotenoids and lycopene [23]. We did not observe the same level of change. The reasons for this are unclear, perhaps due to the small number of vitamin A insufficient or deficient mothers, or perhaps due to the relative amount of moringa leaf powder ingested compared to that used in animal studies being simply lower than what is required to change milk carotenoids. Additionally, several carotenoids trended higher in the intervention group at baseline, though only alpha-carotene was significantly different. This may have masked any small differences that would be seen in a larger cohort.

### 4.2. Limitations

The data are limited by a small sample size, a small number of clusters, and a lower rate of vitamin A deficiency than expected for this rural western Kenyan population. Additionally, the average consumed dose of moringa was less than directed. The study is also limited by a lack of maternal dietary data.

## 5. Conclusions

Moringa’s effect on milk carotenoids was inconclusive in this small population, despite a suggestion that it may increase select carotenoid concentrations such as alpha-carotene. This is the first study to our knowledge to describe changes in human milk retinol and carotenoids with moringa leaf supplementation. Our report adds to the limited body of knowledge on longitudinal changes in human milk carotenoid content and more specific information on milk carotenoids and retinol in a population at risk of vitamin A deficiency. Evaluation in larger samples is warranted.

## Figures and Tables

**Figure 1 nutrients-16-03425-f001:**
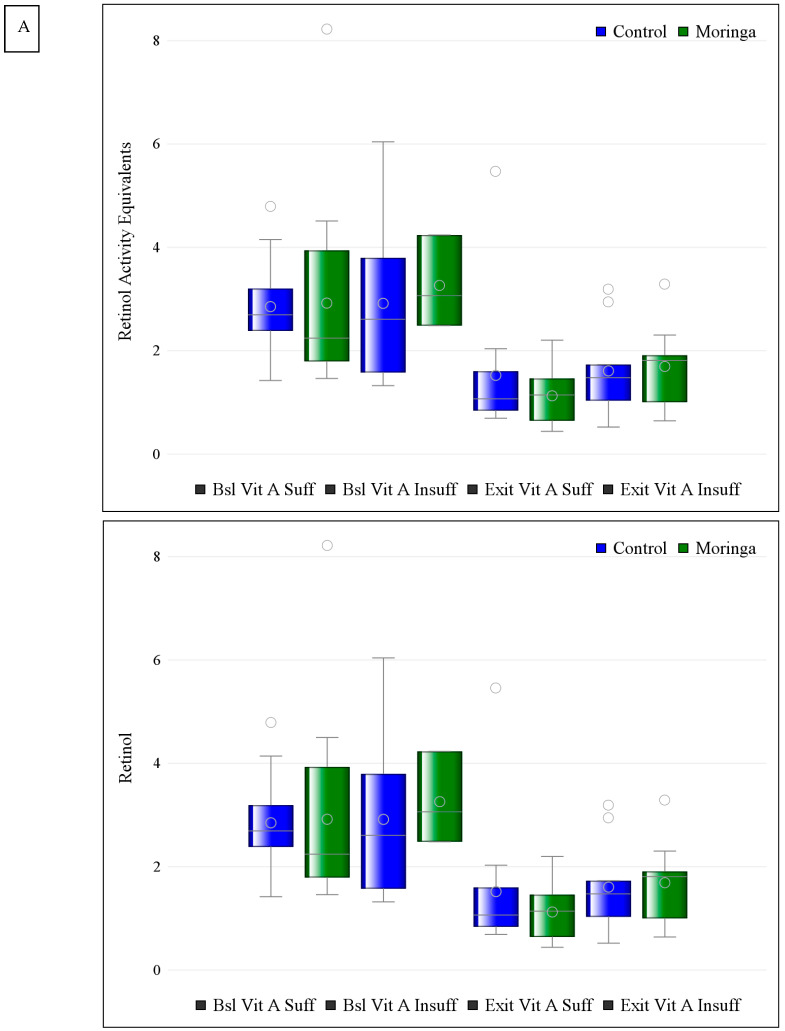
Box plots of milk retinol and select carotenoids according to maternal vitamin A status at baseline (Bsl) and exit with (green) and without (blue) maternal moringa supplementation. Median and IQR are presented. Open circles represent means. Vit A Suff: vitamin A sufficiency defined as retinol binding protein ≥1.05 µg/mL. Vit A Insuff: vitamin A insufficiency defined as serum retinol binding protein <1.05 µg/mL. (**A**) Retinol activity equivalents and retinol; (**B**) alpha-carotene and beta-carotene; (**C**) cis-beta-carotene and total beta-carotene; (**D**) alpha-cryptoxanthin and beta-cryptoxanthin; (**E**) lutein and lycopene; (**F**) cis-lycopene and total lycopene; (**G**) zeaxanthin and total reported carotenoids.

**Table 1 nutrients-16-03425-t001:** Baseline demographics, social, and medical characteristics of mother–infant pairs.

	Control n = 22	Moringa n = 22	*p*-Value
Mothers			
Age at enrollment, y, mean (SE)	28.5 (1.34)	27.4 (1.34)	0.585
Spoke English, n (%)	6 (27)	12 (55)	0.084
Spoke Luo, n (%)	21 (95)	21 (95)	1.000
Spoke Swahili, n (%)	21 (95)	20 (91)	0.568
Prenatal or birth complications, n (%)	1 (5)	0	NE
HIV positive, n (%)	7 (32)	8 (36)	0.758
Additional chronic medical diagnosis, n (%)	0	0	NE
Pre/postnatal vitamin A supplementation, n (%)	0	0	NE
Received food supplements, n (%)	0	0	NE
MUAC at enrollment <23 cm, n (%)	1 (5)	4 (18)	0.199
Infants			
Sex (female), n (%)	10 (45)	14 (64)	0.246
Gestational age, weeks, mean (SE)	38.0 (0.36)	38.0 (0.36)	1.000
Birthweight, g, mean (SE)	3417.6 (99.6)	3486.4 (97.3)	0.624
Spontaneous vaginal delivery, n (%)	18 (82)	22 (100)	NE
WAZ, mean (SE)	0.19 (0.22)	0.29 (0.22)	0.762
LAZ, mean (SE)	0.27 (0.27)	−0.03 (0.27)	0.431
WLZ, mean (SE)	−0.28 (0.37)	0.28 (0.37)	0.299

NE: not estimable, MUAC: mid-upper-arm circumference; WAZ: weight-for-age Z score; LAZ: length-for-age Z score; WLZ: weight-for-length Z score.

**Table 2 nutrients-16-03425-t002:** Serum retinol binding protein and vitamin A status with and without maternal supplementation with moringa leaf powder (20 g/day) for 3 months.

	Baseline			Exit		
Parameter	Control n = 25	Moringa n = 23	*p*-Value	Control n = 23	Moringa n = 23	*p*-Value
Infants						
RBP, µmol/L, LSM (95% CI)	0.673 (0.593, 0.752)	0.694 (0.611, 0.777)	0.716	0.759 (0.572, 0.947)	0.770 (0.582, 0.957)	0.938
Vit A Sufficient, n (%)	1 (4%)	2 (9%)	0.523	3 (13%)	2 (9%)	0.647
Vit A Insufficient, n (%)	24 (96%)	21 (91%)	0.523	20 (87%)	21 (91%)	0.647
Vit A Deficient, n (%)	18 (72%)	12 (52%)	0.176	14 (61%)	11 (48%)	0.391
Mothers						
RBP, µmol/L, LSM (95% CI)	1.409 (1.151, 1.668)	1.680 (1.411, 1.949)	0.151	1.193 (0.966, 1.420)	1.434 (1.212, 1.657)	0.133
Vit A Sufficient, n (%)	16 (64%)	19 (83%)	0.169	12 (52%)	15 (63%)	0.488
Vit A Insufficient, n (%)	9 (36%)	4 (17%)	0.169	11 (48%)	9 (38%)	0.488
Vit A Deficient, n (%)	2 (8%)	2 (9%)	0.932	3 (13%)	1 (4%)	0.313

RBP, retinol binding protein. Vit A, vitamin A. Vitamin A status defined as sufficient for serum RBP > 1.05 µmol/L, insufficient for serum RBP < 1.05 µmol/L, and deficient for serum RBP < 0.07 µmol/L at that timepoint (baseline or exit). The back-transformed least squares means estimate is presented along with the back-transformed 95% confidence interval.

**Table 3 nutrients-16-03425-t003:** Retinol and carotenoid concentrations of milk produced by women with and without maternal supplementation with moringa leaf powder (20 g/day) for 3 months.

	Baseline LSM (95% CI), µg/mL			Exit LSM (95% CI), µg/mL		
Parameter	Control n = 22	Moringa n = 22	*p*-Value	Control n = 22	Moringa n = 22	*p*-Value
RAE ^†^	2.80 (2.28, 3.35)	2.80 (2.28, 3.36)	0.989	1.49 (1.14, 1.86)	1.32 (0.98, 1.68)	0.509
Retinol	2.79 (2.27, 3.35)	2.80 (2.28, 3.35)	0.989	1.49 (1.14, 1.86)	1.32 (0.98, 1.68)	0.509
**Provitamin A carotenoids**						
alpha-Carotene	0.004 (0.002, 0.007)	0.008 (0.006, 0.011)	0.024	0.002 (0.001, 0.003)	0.003 (0.003, 0.004)	0.014
beta-Carotene	0.020 (0.011, 0.030)	0.031 (0.022, 0.041)	0.118	0.017 (0.011, 0.023)	0.016 (0.010, 0.022)	0.885
cis-beta-Carotene	0.006 (0.004, 0.008)	0.008 (0.006, 0.010)	0.299	0.005 (0.004, 0.006)	0.004 (0.003, 0.005)	0.554
Total beta-Carotene	0.026 (0.015, 0.038)	0.039 (0.027, 0.051)	0.148	0.021 (0.014, 0.028)	0.020 (0.013, 0.027)	0.805
beta-Cryptoxanthin	0.049 (0.031, 0.067)	0.036 (0.018, 0.054)	0.321	0.018 (0.011, 0.025)	0.009 (0.002, 0.016)	0.085
**Other Carotenoids**						
alpha-Cryptoxanthin	0.007 (0.004, 0.010)	0.011 (0.008, 0.014)	0.063	0.004 (0.003, 0.005)	0.005 (0.004, 0.005)	0.537
Lutein	0.154 (0.090, 0.218)	0.239 (0.174, 0.305)	0.067	0.133 (0.081, 0.185)	0.166 (0.114, 0.218)	0.377
Lycopene	0.015 (0.009, 0.021)	0.022 (0.015, 0.028)	0.145	0.011 (0.009, 0.013)	0.010 (0.008, 0.012)	0.498
cis-Lycopene	0.016 (0.008, 0.024)	0.023 (0.015, 0.031)	0.242	0.010 (0.008, 0.012)	0.009 (0.007, 0.011)	0.408
Total Lycopene	0.031 (0.016, 0.046)	0.045 (0.030, 0.059)	0.189	0.021 (0.017, 0.025)	0.019 (0.015, 0.023)	0.428
Zeaxanthin	0.037 (0.020, 0.053)	0.044 (0.028, 0.061)	0.519	0.017 (0.006, 0.028)	0.028 (0.017, 0.039)	0.170
Total Reported Carotenoids	0.306 (0.199, 0.416)	0.416 (0.307, 0.528)	0.158	0.213 (0.144, 0.283)	0.248 (0.179, 0.319)	0.473

^†^ RAE, retinol activity equivalents. RAE is calculated as retinol + (β-carotene/12) + (α-carotene/24) + (β-cryptoxanthin/24) using LSM levels [28]. LSM, least squares means. The back-transformed LSM estimate is presented along with the back-transformed 95% confidence interval.

**Table 4 nutrients-16-03425-t004:** Retinol and carotenoid concentrations of milk produced by women with vitamin A insufficiency at baseline.

	Baseline LSM (95% CI), µg/mL			Exit LSM (95% CI), µg/mL		
Parameter	Control n = 8	Moringa n = 3	*p*-Value	Controln = 8	Moringa n = 3	*p*-Value
RAE ^†^	2.78 (1.75, 3.97)	3.23 (1.53, 5.38)	0.653	1.36 (0.78, 1.98)	1.70 (0.74, 2.81)	0.528
Retinol	2.78 (1.75, 3.97)	3.23 (1.52, 5.37)	0.656	1.35 (0.78, 1.98)	1.70 (0.74, 2.81)	0.529
**Provitamin A carotenoids**						
alpha-Carotene	0.004 (0.000, 0.008)	0.009 (0.002, 0.016)	0.165	0.002 (0.000, 0.004)	0.006 (0.003, 0.009)	0.027
beta-Carotene	0.014 (0.002, 0.027)	0.038 (0.017, 0.058)	0.055	0.013 (0.005, 0.021)	0.024 (0.011, 0.037)	0.126
cis-beta-Carotene	0.005 (0.002, 0.008)	0.009 (0.004, 0.014)	0.106	0.004 (0.002, 0.005)	0.005 (0.003, 0.008)	0.215
Total beta-Carotene	0.019 (0.003, 0.034)	0.047 (0.022, 0.072)	0.060	0.016 (0.007, 0.025)	0.029 (0.014, 0.045)	0.129
beta-Cryptoxanthin	0.020 (0.011, 0.029)	0.062 (0.047, 0.077)	<0.001	0.007 (0.004, 0.010)	0.016 (0.011, 0.020)	0.005
**Other Carotenoids**						
alpha-Cryptoxanthin	0.006 (0.003, 0.009)	0.016 (0.011, 0.021)	0.004	0.004 (0.003, 0.005)	0.006 (0.004, 0.008)	0.089
Lutein	0.140 (0.057, 0.225)	0.322 (0.183, 0.466)	0.033	0.116 (0.015, 0.219)	0.266 (0.098, 0.440)	0.120
Lycopene	0.014 (0.007, 0.021)	0.017 (0.006, 0.029)	0.612	0.011 (0.008, 0.014)	0.011 (0.007, 0.015)	0.958
cis-Lycopene	0.015 (0.005, 0.025)	0.019 (0.003, 0.035)	0.622	0.010 (0.007, 0.013)	0.010 (0.004, 0.016)	0.910
Total Lycopene	0.029 (0.011, 0.046)	0.037 (0.008, 0.065)	0.613	0.021 (0.015, 0.027)	0.021 (0.011, 0.030)	0.987
Zeaxanthin	0.026 (−0.009, 0.061)	0.105 (0.048, 0.163)	0.025	0.016 (0.002, 0.031)	0.044 (0.020, 0.068)	0.052
Total Reported Carotenoids	0.241 (0.129, 0.356)	0.597 (0.403, 0.799)	0.006	0.179 (0.053, 0.308)	0.387 (0.176, 0.608)	0.090

^†^ RAE, retinol activity equivalents. RAE is calculated as retinol + (β-carotene/12) + (α-carotene/24) + (β-cryptoxanthin/24) using least squares means levels [28]. LSM, least squares means. The back-transformed LSM estimate is presented along with the back-transformed 95% confidence interval. Vitamin A insufficiency is defined as serum retinol binding protein <1.05 µg/L.

**Table 5 nutrients-16-03425-t005:** Retinol and carotenoid concentrations of milk produced by women with vitamin A sufficiency at baseline.

	Baseline LSM (95% CI), µg/mL			Exit LSM (95% CI), µg/mL		
Parameter	Control n = 14	Moringa n = 18	*p*-Value	Controln = 14	Moringa n = 18	*p*-Value
RAE ^†^	2.80 (2.15, 3.52)	2.78 (2.20, 3.40)	0.959	1.56 (1.09, 2.07)	1.27 (0.88, 1.70)	0.359
Retinol	2.80 (2.15, 3.51)	2.78 (2.20, 3.40)	0.960	1.56 (1.09, 2.07)	1.27 (0.88, 1.70)	0.359
**Provitamin A carotenoids**						
alpha-Carotene	0.004 (0.001, 0.008)	0.008 (0.005, 0.011)	0.082	0.002 (0.001, 0.003)	0.003 (0.002, 0.004)	0.090
beta-Carotene	0.024 (0.011, 0.037)	0.031 (0.020, 0.043)	0.402	0.019 (0.011, 0.028)	0.015 (0.008, 0.023)	0.472
Total beta-Carotene	0.031 (0.015, 0.047)	0.039 (0.025, 0.054)	0.453	0.024 (0.014, 0.034)	0.019 (0.010, 0.027)	0.411
cis-beta-Carotene	0.007 (0.004, 0.010)	0.008 (0.005, 0.011)	0.636	0.005 (0.004, 0.006)	0.004 (0.003, 0.005)	0.245
beta-Cryptoxanthin	0.066 (0.041, 0.090)	0.033 (0.012, 0.055)	0.052	0.024 (0.014, 0.034)	0.008 (−0.000, 0.017)	0.018
**Other Carotenoids**						
alpha-Cryptoxanthin	0.008 (0.004, 0.012)	0.010 (0.007, 0.014)	0.329	0.000 (−0.002, 0.003)	−0.001 (−0.003, 0.002)	0.473
Lutein	0.161 (0.074, 0.250)	0.235 (0.157, 0.315)	0.212	0.143 (0.078, 0.208)	0.151 (0.094, 0.209)	0.843
Lycopene	0.015 (0.006, 0.024)	0.023 (0.015, 0.031)	0.205	0.011 (0.008, 0.014)	0.010 (0.008, 0.012)	0.587
cis-Lycopene	0.017 (0.005, 0.029)	0.024 (0.014, 0.035)	0.333	0.010 (0.008, 0.013)	0.009 (0.007, 0.011)	0.478
Total Lycopene	0.032 (0.012, 0.053)	0.048 (0.029, 0.066)	0.263	0.021 (0.016, 0.026)	0.019 (0.014, 0.023)	0.514
Zeaxanthin	0.043 (0.025, 0.061)	0.036 (0.020, 0.052)	0.568	0.017 (0.001, 0.033)	0.026 (0.012, 0.040)	0.417
Total Reported Carotenoids	0.344 (0.196, 0.496)	0.404 (0.272, 0.539)	0.545	0.232 (0.144, 0.322)	0.229 (0.151, 0.308)	0.954

^†^ RAE, retinol activity equivalents. RAE is calculated as retinol + (β-carotene/12) + (α-carotene/24) + (β-cryptoxanthin/24) using least squares means levels [28]. LSM, least squares means. The back-transformed LSM estimate is presented along with the back-transformed 95% confidence interval. Vitamin A sufficiency is defined as serum retinol binding protein ≥1.05 µg/L.

## Data Availability

Data is available upon request to the corresponding author.

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
