# Peer review of "Impact of Maternal Moringa oleifera Leaf Supplementation on Milk and Serum Vitamin A and Carotenoid Concentrations in a Cohort of Breastfeeding Kenyan Women and Their Infants"

_nutrients, 2024, doi:10.3390/nu16193425_

Round 1

Reviewer 1 Report

Comments and Suggestions for Authors

The manuscript “Impact of maternal Moringa oleifera leaf supplementation on 1 milk and serum vitamin A and carotenoid concentrations in a 2 cohort of breastfeeding Kenyan women and their infants” by Suzanna L. Attia (corresponding author), Silvia A. Odhiambo,  Jerusha N. Mogaka, Raphael Ondondo,  Aric 5 Schadler, Kristen McQuerry,  George J. Fuchs and III,  Janet E. Williams, 6Michelle K. McGuire, Carrie Waterman,  Kerry Schulze, and Patrick M. Owuor is interesting, however, I have some doubts.

1.       There is nothing about the diet of the mothers. The porridge twice a day is not a complete daily nutrition. There should be a query about diet habits and estimating vitamin A and its precursors in daily intake. – see also point 2 below.

2.       It is a pity, that the authors did not measure the vitamin A precursors in used moringa powder. In the manuscript are data from other studies – “Moringa leaves contain high amounts of pro-vitamin A carotenoids, especially beta-carotene and beta-cryptoxanthin, which  some studies estimate as up to 36.4 μg of retinol activity equivalents (RAE) per gram dried moringa leaf powder [18], [19]” (unfortunately I am not able to find these papers). The RAE content is probably region and season-sensitive. I have found that in moringa powdered leaves there is only 12 ug RAE/g  - Improving Blood Retinol Concentrations with Complementary Foods Fortified with Moringa oleifera Leaf Powder – A Pilot Study By Laurene Boateng, Irene Ashley, Agartha Ohemeng, Matilda Asante, and Matilda Steiner-Asiedua. Yale J Biol Med. 2018 Jun; 91(2): 83–94.

Short calculation - 12 ug RAE/g x 20 g (effective 12-14 g/day – “moringa leaf powder use was 11.8 ± 3.8 g/day between months 1 and 2 and 14.9 ± 3.4 g/day between months 2 and 3 of the 3-month study”) = 240 ug RAE expected and 144-168 effective. It is about 11-13% of the recommended daily intake of RAE by lactating women. (1300 ug/day; Institute of Medicine. Food and Nutrition Board. Dietary Reference Intakes for Vitamin A, Vitamin K, Arsenic, Boron, Chromium, Copper, Iodine, Iron, Manganese, Molybdenum, Nickel, Silicon, Vanadium, and Zinc. Washington, DC: National Academy Press; 2001. So the dose is small and the effect is also small as reported. It could be expected.

3.       I do not understand – samples of milk “stored on dry ice, then at -2C then at - 80C until analysis”. They were firstly deeply frozen (dry ice generates about -78oC) later warmed to -2oC and frozen deeply once again. Why?  

4.       “heel for infant with up to 2mL collected/sample” – First – WHO recommends venous samples in infants. Secondly, 2 ml is a huge volume – to be true, it is impossible to obtain such a volume from infants from capillary puncture.

5. Used statistics is not clear, at least description is not clear.

6.       The Limitations para is usually at the end of the Discussion.

7.       The conclusions except the first phrase are not conclusions.  

Reviewer 2 Report

Comments and Suggestions for Authors

I had to revise the article entitled ” Impact of maternal Moringa oleifera leaf supplementation on 1 milk and serum vitamin A and carotenoid concentrations in a 2 cohort of breastfeeding Kenyan women and their infants 3” submitted for publishing in Nutrients journal.

The present study investigated the role of maternal moringa supplementation in improving vitamin A status in both mother and children if this supplementation is given to mothers. It is a well conducted study but several improvements are needed in order to be accepted for publication

1.     Please revise tables 1 to 3. I cannot analyze the data from  these tables

2.     The authors should explain why they decided to take milk samples after breastfeeding and not before.

3.     In table 4 and figures there is a duplication of the results. The authors should choose what method they prefer to present the same results: figures or tables.

4.     In discussion section first paragraph should be excluded. The advantages of moringa leaves were already presented in introduction section. I suggest presenting briefly the main findings here. I suggest structuring the discussion section presenting first the main findings, analyzing them in comparison to other previous research and then the limitations

5.     The entire article needs a typing revision in order to use the same font.

Round 2

Reviewer 1 Report

Comments and Suggestions for Authors

1. I would suggest to add to the limitations that the consumption of leaf powder was lower than recommended. Also, you do not add the estimated RAE intake in Moringa powder as a percentage of the recommended RAE intake by WHO for lactating women

2. The point mentioned below was not corrected as stated in the answers

samples of milk “stored on dry ice, then at -2C then at - 80C until analysis”.
They were firstly deeply frozen (dry ice generates about -78oC) later warmed to -2oC and frozen deeply once again. Why?
-Thank you for catching this. It was -20C. We have corrected this. Dry ice was very brief and standard for sample transport. -20C was adequate for the time period of storage used.

I am pediatrician and I am sure, that it is impossible to possess 2 ml of blood using a heel puncture. What is more: Venepuncture is the preferred method of blood sampling for term neonates, and causes less pain than heel-pricks (64). WHO Guidelines on Drawing Blood: Best Practices in Phlebotomy.

Author Response

  1. I would suggest to add to the limitations that the consumption of leaf powder was lower than recommended. Also, you do not add the estimated RAE intake in Moringa powder as a percentage of the recommended RAE intake by WHO for lactating women. 
    1. Response: thank you again for your comments. We have added more background regarding the use of postpartum Vit A supplementation. We have added more on the RAE provision in moringa to the Discussion. 

2. The point mentioned below was not corrected as stated in the answers

samples of milk “stored on dry ice, then at -2C then at - 80C until analysis”.
They were firstly deeply frozen (dry ice generates about -78oC) later warmed to -2oC and frozen deeply once again. Why?
-old response: Thank you for catching this. It was -20C. We have corrected this. Dry ice was very brief and standard for sample transport. -20C was adequate for the time period of storage used.

- Response: thank you, we apologize for the error and have corrected it again. This times the correction seems to have stuck. 

I am pediatrician and I am sure, that it is impossible to possess 2 ml of blood using a heel puncture. What is more: Venepuncture is the preferred method of blood sampling for term neonates, and causes less pain than heel-pricks (64). WHO Guidelines on Drawing Blood: Best Practices in Phlebotomy.

- Response: thank you for the reference. We used the protocol from our own neonatal intensive care unit of the University of Kentucky in consultation with our own team and consultants of Kenyan medical practitioners. We understand that practices may vary in different places. The maximum taken was indeed 2ml.  

Reviewer 2 Report

Comments and Suggestions for Authors

I have an observation regarding the format of the table which are not completely visible. Table 4 and 5 contain the same data showed in the figures. I suggest to keep only one form of the presentation.

Author Response

I have an observation regarding the format of the table which are not completely visible. Table 4 and 5 contain the same data showed in the figures. I suggest to keep only one form of the presentation.

- Response. Thank you again for your review. We mentioned to the editors and in our prior response that the tables are in Layout format. For this reason when one downloads the mansucripts, they appear to be cut off, however in print they should be in layout format and take up the whole page in 90 degrees rotation from what you are seeing now. The tables and figures actually represent different information. Thank you for your time.